# Propagation of viral bioaerosols indoors

**Olga B. Kudryashova**[¤]*, **Evgeny V. Muravlev, Aleksandra A. Antonnikova, Sergey S. Titov**

Institute for Problems of Chemical and Energetic Technologies, SB RAS, Biysk, Russia

¤ Current address: IPCET SB RAS, Biysk, Altai Krai, Russia
* olgakudr@inbox.ru

## Abstract

Here we look into the spread of aerosols indoors that may potentially carry viruses. Many viruses, including the novel SARS-CoV-2, are known to spread via airborne and air-dust pathways. From the literature data and our research on the propagation of fine aerosols, we simulate herein the carryover of viral aerosols in indoor air. We demonstrate that a lot of fine droplets released from an infected person's coughing, sneezing, or talking propagate very fast and for large distances indoors, as well as bend around obstacles, lift up and down over staircases, and so on. This study suggests equations to evaluate the concentration of those droplets, depending on time and distance from the source of infection. Estimates are given for the safe distance to the source of infection, and available methods for neutralizing viral aerosols indoors are considered.

## Introduction

The coronavirus pandemic that stroke the world in 2020 has posed a series of questions for humanity that need to be solved. And these are not only medicine-related issues but also those relating to other specialists. In particular, of importance is research on generation, evolution, and propagation of bioaerosols containing viral particles. The novel coronavirus infection is communicated not only via contact but also via airborne pathway [1]. This is also corroborated by the reported results Liu et al. [2]. Liu et al. detected a plenty of RNA fragments of the novel coronavirus inside Wuhan hospitals; moreover, not only were they detected in the wards with coronavirus-infected patients but also in other sections of the hospitals to which the virus should not get by any means. The genomic fragments of SARS-CoV-2 were found both in fine and coarse aerosols particles; however, they were mostly found in 0.25–0.5 μm droplets.

The same tendency was noticed for droplets carrying flu viruses [3]. It turned out that aerosol particles with a size below 5 μm contained 8.8. times more replicas than larger particles. This speaks of fine respiratory aerosol particles as being more dangerous in terms of potential contagion. Besides, fine aerosol particles may sustain in the air and penetrate deeper into the lungs, causing more severe progression of the disease [4, 5]. Perhaps the concentration of viral particles is not so high as to cause a danger of infection, but the very fact of the transfer of viral particles by small aerosol droplets far from the source of origin is interesting.

**Data Availability Statement:** All relevant data are within the manuscript.

**Funding:** The work was supported by the Russian Ministry of Education and Science of the Russian Federation (mon.gov.ru) grant (state task). The funder had no role in study design, data col-lection

and analysis, decision to publish, or preparation of the manuscript.

**Competing interests:** The authors have declared that no competing interests exist.

Knowing the propagation regularities of viral bioaerosols would help map out ways of preventing dangerous infections. Relatively large respiratory particles (above 15 μm in size) follow the ballistic trajectory and settle on the surface in close proximity to the source of infection. Ordinary medical masks, except for paper ones, can efficiently hold them up [6–9].

However, smaller droplets can be carried over even farther by the airflow [10], in which case small aerosol particles are potential virus-carriers and foster the pandemic spread because they are yet big enough to contain thousands of viral and bacterial pathogenic microorganisms [11]. Asadi et al. [12] assert that fine aerosols (with a particle size below 5 μm) exhaled with usual speech serve as a crucial medium for the transmission of SARS-CoV-2.

It should be noted that water particles less than 5 μm would have evaporated very fast (within a few seconds at a typical low-level air humidity indoors). But this is not the case for particles of saliva and mucus in which surfactants are contained. These surfactants diminish the evaporation rate of droplets by a factor of 2–10 (subject to the air humidity) [13]. On the other hand, these substances protect viruses inside a droplet, extending their survivability. van Doremalen N et al. [14] demonstrated that the SARS-CoV-2 virus can stay alive as an aerosol for at least 3 h. However, high temperature (above $t_C \sim 30 \pm 2.4\ ^0C$ [15]) and humidity appear to reduce the viability and transmissibility of COVID-19 [16].

By now, it is thought for sure that bioaerosols resulting from coughing, sneezing, and talking are less dangerous in outdoor space than indoors [17]. The major bulk of dangerous aerosol particles settle down within the 'social distance' of 1–2 m if there is no wind or wind speed is at most 4 km/h. A finer fraction of bioaerosols scatters rapidly to safe concentrations in outdoor space.

The situation is different in indoor space, even if people keep the social distance from each other in indoor rooms and the source of infection persists over a long time (an infected or asymptomatic person). This source of infection when coughing, sneezing, or talking releases the tiniest virus-bearing saliva particles into the air that may hover in the air for a long time without settling and evaporating [18].

Without special air purification and disinfection tools, the particle concentration of a bioaerosol issued from the source of infection will only be growing if the infected person is inside the room. Tiny micron-sized particles can spread fast in a room in spite of obstacles. As the convincing 3D-model simulation demonstrated [19], most of the droplets of respiratory origin reside in the air for quite a long time and propagate rapidly and a long way from the source. The simulation also showed that usual ventilation does not reduce the particle concentration. The risk of accumulated critical viral exposure may persist for a few minutes after a cough at a distance of about up to 4 m from the source. Vuorinen et al. [19] place attention on the risks associated with crowded indoor spaces (offices, schools, public transport).

We investigated earlier the propagation of fine aerosols with a typical particle size below 15 μm in space and derived approximate equations that can estimate the diffusive propagation velocity of those aerosols in confined space [20]. Here we used and elaborated the earlier obtained results.

The present study aimed to describe the indoor propagation of a respiratory aerosol cloud as a potential virus-carrier. We are going demonstrate experimentally and theoretically that a lot of fine droplets released from an infected person's coughing, sneezing, or talking propagate very fast and for large distances indoors, as well as bend around obstacles, lift up and down over staircases, and so on.

## Materials and methods

### Initial data for simulation

Chao et al. [21] measured the sizes of particles resulting from coughing, sneezing, and loud talking by using optical techniques based on the Mie scattering theory. The average geometric

diameter of the particles was estimated to be 13.5–16.0 μm. Yang et al. [22] estimated the particle size at coughing to be between 0.62 and 15.9 μm, with the mean diameter being 8.35 μm. The initial ejection velocity of droplets at sneezing was measured by Scharfman et al. [23] and reached 35 m/s.

## Methods

We applied physical simulation of the propagation of a fine aerosol ejected in confined space of different configurations. Two pulsed generators were employed as the source of aerosol [20]. The representative sizes of droplets were approximately within the range of interest. The particle diameter ranged from 0.5 to 20 μm with a distribution mode of 10 μm. The generation and ejection velocities of the droplets were an order of magnitude greater than those at coughing and sneezing. But, as is shown hereinafter, the initial ejection velocity of the droplets does not affect the final travel velocity and distance of the particles due to air drag. The liquid to be atomized in the experiment weighed 6 mL (3 mL per each generator).

## Materials

We used a 20 wt.% aqueous solution of glycerol as the model liquid. Water droplets of micron sizes at typical air humidity and temperature would evaporate indoors within a few seconds, whereas the model solution droplets would not evaporate completely for a long time. Bioaerosol droplets containing saliva, mucus, and epithelium particles would behave the same. The evaporation time of these particles, is 2–10 times greater than that of water, as reported Vejerano et al. [13]. Furthermore, once evaporated, the liquid leaves a dry residue. For our model solution, the dry residue is imitated by an almost non-vaporizable proportion of glycerol.

By setting the problem, we did not consider the evaporation stage of the volatile fraction of particle matter, which decreases the sizes of the particles. Under low humidity conditions typical of indoor environments, the evaporation stage ends rapidly. This is followed by a slower stage of the diffusive propagation of non-volatile residues of droplets. It is the sizes of these droplets that we measured by instruments, and it is the propagation of those droplets in confined space that we investigated herein.

The concentration and size of aerosol particles were measured by the small-angle scattering method using a stand-off laser unit [24]. Measurements are available in S1 Table.

We simulated the aerosol propagation in space of complex configurations. In the first option, a test box in two configurations was positioned horizontally at the same level. The space configurations and the location of aerosol generators and instruments are illustrated in Fig 1.

In the second option, an experimental space in two configurations imitated a staircase. In this case, the boxes were oriented vertically, while the generators were positioned either on the bottom or on the top (Fig 2).

## Mathematical model

Consider the propagation of a cloud of particles with representative diameter $d$ in confined space in the air at rest. The particles are ejected from the point source with initial velocity $u_0$. In this case, they get retarded while traveling in the air and come to a stop at some distance $R$ during time $t_0$. A spherical cloud with radius $R$ is formed. Further propagation of the particles is due to diffusion.

The whole process can be classified into two phases: a short-term phase when particles scatter to generate a cloud with radius $R$, and a long-term phase of diffusive propagation. In the second phase, some largest particles will settle down due to gravity.

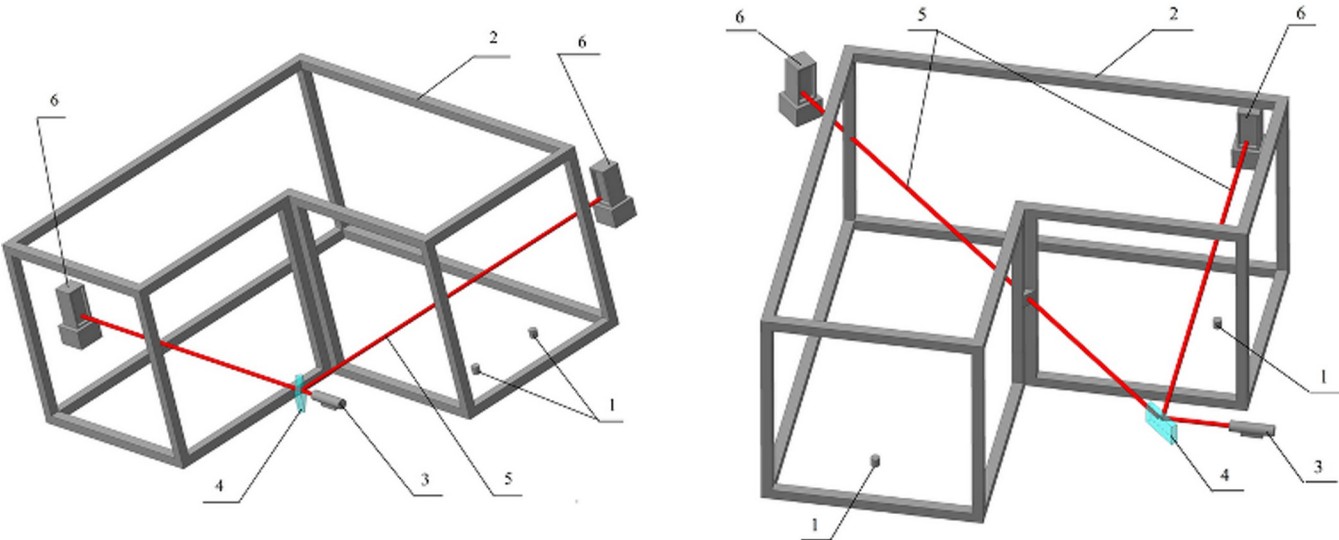

**Fig 1. The horizontal configuration and location of generators in the experiment.** (1) atomizers, (2) box walls, (3) laser, (4) mirror, (5) laser beam, and (6) photodetectors. (A) L-shaped configuration, atomization at one spot. (B) L-shaped configuration, atomization at two spots.

**Primary scatter and formation time of fine aerosol cloud.**   The equation for particle travel is written as:

$$\frac{\pi \rho_p d^3}{3}\frac{du}{dt} = -\pi d^2 C_D \frac{\rho u^2}{4},$$ (1)

where $\rho_p$, $\rho$ are the densities of the droplet and air, respectively; $u$ is the particle velocity; $C_D = 24/\text{Re}$ is the dimensionless drag coefficient; and $Re = \frac{d\rho u}{\mu}$ is the Reynolds number. The integral of Eq (1):

$$u = u_0 \exp\left(-\frac{18\mu}{d^2 \rho_p}t\right),$$ (2)

where $u_0$ is the initial particle velocity.

Distance $r(t)$ the particle has passed in the air is defined by the integration of Eq (2). The particle gets retarded in the air, reaching asymptotically maximum distance $R$ within a time $t_0$ (Fig 3). In our model case, the initial particle velocity was $u_0 = 300$ m s$^{-1}$, the cloud radius was $R = 21$ cm, and the cloud formation time was $t_0 = 2.6$ ms.

Maximum distance $R$ is linearly dependent on initial velocity $u$ (Fig 4). The time, over which the maximum distance is covered, is estimated at several milliseconds.

At an ejection velocity consistent with the data [23] for coughing, small particles will stop at a distance less than 3 cm from the source. At an ejection velocity of $u = 300$ m/s small particles will stop at a distance about 20 cm from the source. Thus, that difference in initial velocity is not important for describing the propagation of particles over distances of several meters.

**Diffusive propagation of particles in space.**   In spherical approximation, the problem of diffusion of aerosol particles can be written as follows:

$$\frac{\partial c}{\partial t} = \frac{D}{r^2}\frac{\partial}{\partial r}\left(r^2 \frac{\partial c}{\partial r}\right),$$ (3)

with initial and boundary conditions:

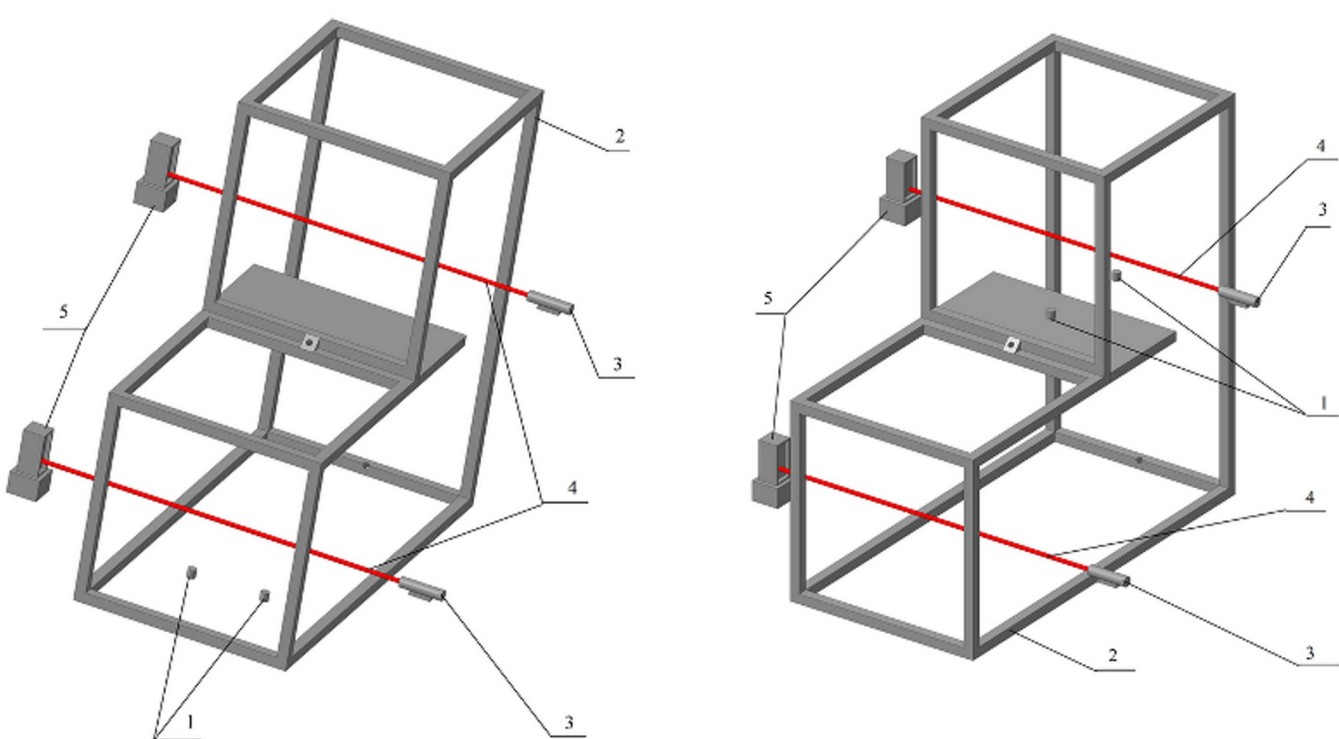

**Fig 2. Schematics of experimental configurations imitating a staircase.** (A) generators on the bottom and (B) generators on the top. (1) atomizers, (2) box walls, (3) laser, (4) laser beam, and (5) photodetectors.

$$t = 0, c = c_0 \text{ at } r < R; c = 0 \text{ at } r > R,$$

$$r = 0 : \quad \frac{\partial c}{\partial r} = 0; \quad r \to \infty : \quad \frac{\partial c}{\partial r} = 0,$$

where $c$ is the mass concentration of particles; $c_0$ is the mass concentration in the primary cloud having radius $R$ which is calculated by solving the problem of initial scatter of droplets; $c_0 = \frac{3m_0}{4\pi R^3}$, $r$ is the radial coordinate; $t$ is the time; $D$ is the effective coefficient of diffusion; and $m_0$ is the total weight of the liquid.

The solution to the diffusion equation is written as:

$$C = \frac{1}{2} \left\{ \text{erfc}\left(\frac{r - R}{2\sqrt{Dt}}\right) - \frac{2}{r}\sqrt{\frac{Dt}{\pi}}\exp\left[-\frac{(r - R)^2}{4Dt}\right] - \text{erfc}\left(\frac{r + R}{2\sqrt{Dt}}\right) + \frac{2}{r}\sqrt{\frac{Dt}{\pi}}\exp\left[-\frac{(r + R)^2}{4Dt}\right] \right\}, (4)$$

where $C(r,t) = c/c_0$ is the dimensionless concentration.

**On the effective coefficient of diffusion.** For particles with $d = 10$ μm, the Brownian diffusion coefficient is $D_b \sim 10^{-12}$ m² s⁻¹. At this diffusion coefficient value, the aerosol cloud would have not spread noticeably from the source within about a few hours. This contradicts the results from numerous studies. Fine particles propagate much faster and farther from the source than the Brownian diffusion would have allowed, even if there is no forced ventilation. Air microflows caused by convection are always present in a room. They entrain fine aerosol particles easily.

The effective diffusion coefficient of particles having $d = 10$ μm was experimentally determined by comparing calculated and measured particle concentrations at different distances from the pulsed aerosol generator [20]. The calculated effective diffusion coefficient was

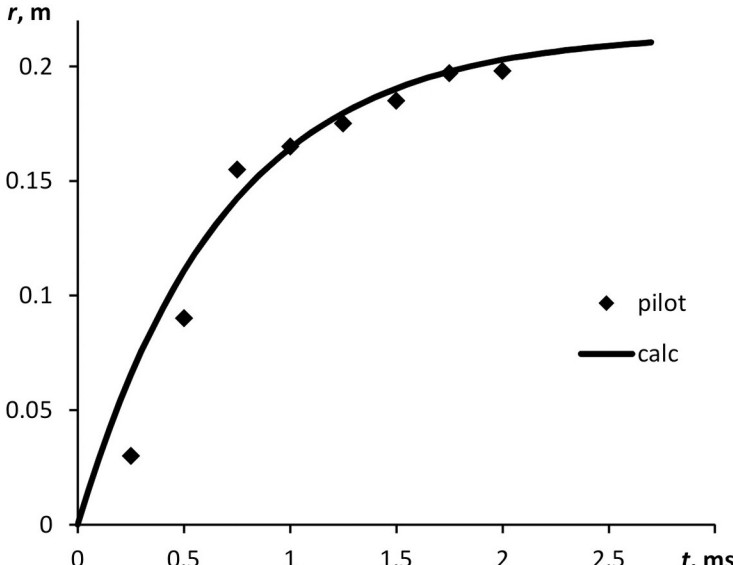

**Fig 3. Distance passed by a water droplet of 15 μm in diameter when it gets retarded in the air. Initial velocity $u$ = 300 m/s.** Experimental dots were obtained by high-speed video recording; the curve stands for calculation.

$D = 1.6 \cdot 10^{-3}$ m$^2$ s$^{-1}$. This is roughly $10^6$ times as great as the Brownian diffusion coefficient for particles of the same sizes. In this study, we used that value of the diffusion coefficient.

## Results

### Experiment with aerosol propagation in a box of complex configuration

The experimental results for the box of horizontal configurations (Fig 1) are illustrated in Fig 5. The aerosol distribution rapidly became almost uniform within 3–5 min in small space

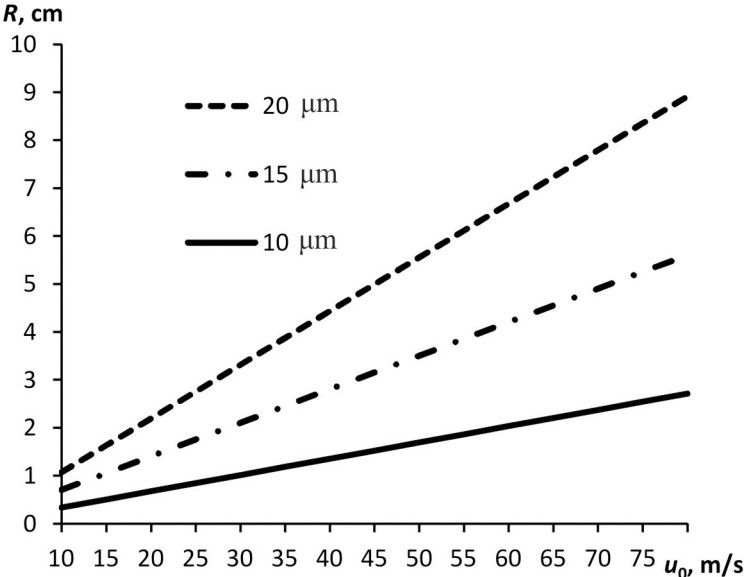

**Fig 4. Distance of retardation of a water droplet with different diameters plotted against initial velocity.**

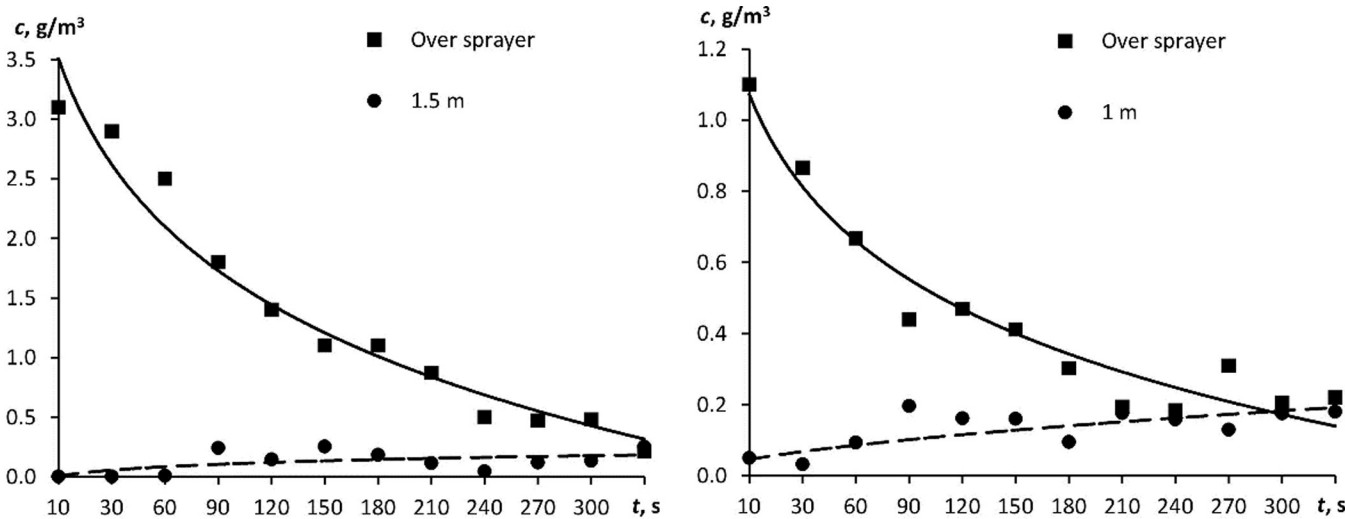

**Fig 5. A time course of aerosol particle concentration for the test box of horizontal configuration (Fig 1).** (A) L-shaped configuration, atomization at one spot; (B) L-shaped configuration, atomization at two spots.

(typical dimensions of 1–2 m), as demonstrated by both the calculations and the experiment. The aerosol particle concentration was lower than expected (6 g was atomized in a 3 m³ volume; expected uniform concentration was 2 g m⁻³): 0.2–0.3 g m⁻³. This was due to the droplets evaporating and depositing on the walls.

Immediately 3–5 min after atomization, all the experimental dots lay in the same region of values (within a measurement error). For the configuration with two atomizers located at the same spot (Fig 1A), the initial particle concentration above the atomizers (0.2 cm away from the source) was higher than that for the configuration with two atomizers located at different spots (Fig 1B). In the most distant corner (1.5 m away from the source), the aerosol particle concentration was leveling off somewhat longer, but in 5 min, it also became approximately equal to 0.2 g m⁻³. The uniform concentration then remained almost unchanged for a few hours.

Fig 6 displays the variation in aerosol particle concentration when the aerosol propagated vertically (staircase imitation, Fig 2). When the aerosol propagated vertically, the concentration was leveling off for 3–5 min, as was in the case of horizontal propagation. This suggests that the assumption of the same diffusion coefficient value, irrespective of the gravitational vector, was valid. Fig 6B depicts the calculated variation in aerosol particle concentration above the atomizer (0.2 m away from the source) at distances of 1 and 1.5 m from the source when 6 g of aerosol was atomized. The calculated curves also demonstrate that the aerosol concentration in space levels off for a few minutes.

## Assessment of characteristic velocities of processes

According to Einstein's law, squared particle displacement $h$ due to diffusion is proportional to the observation time:

$$h^2 = 2Dt, \tag{5}$$

hence, knowing the diffusion coefficient value, one can estimate the time of particle

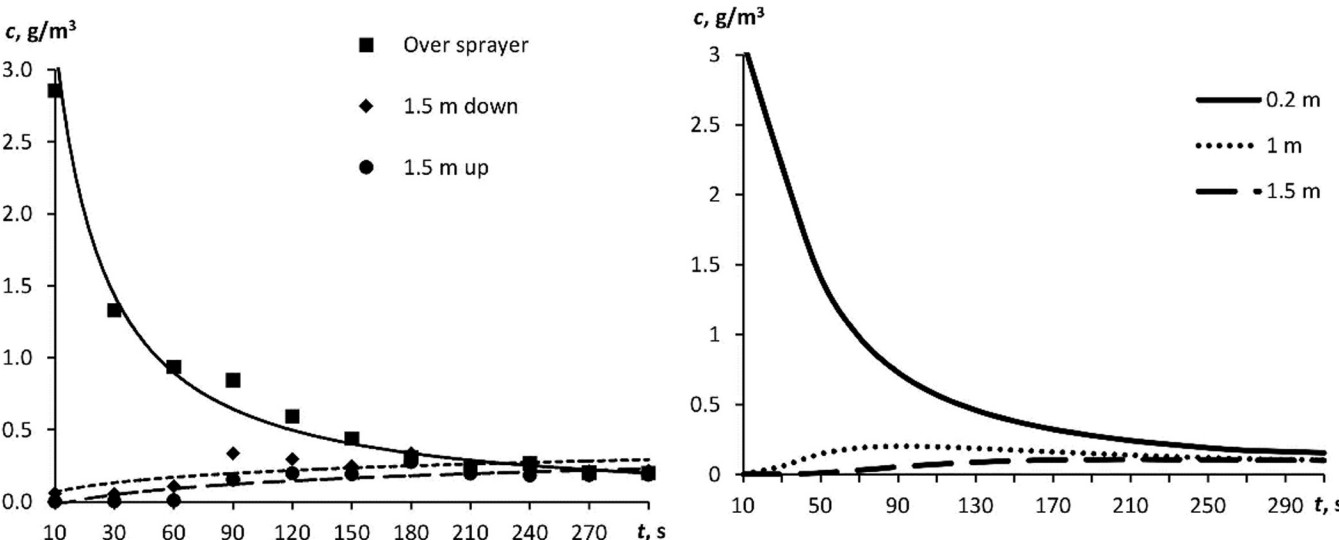

**Fig 6. A time course of aerosol particle concentration.** (A) the experiment in the test box of vertical configuration (Fig 2) and (B) calculation for distances of 0.2, 1, and 1.5 m from the source, independent of configuration.

displacement to a specified distance:

$$t_d = \frac{h^2}{2D}. \tag{6}$$

The diffusion velocity depends on the distance from the source but does not on the particle sizes:

$$u_d = \frac{h}{t_d} = \frac{2D}{h}. \tag{7}$$

In most countries, the common social distance is 1–2 m. As per Eq (6), the time of propagation of a fine aerosol particle to that distance will hence be 5–20 min. Within that time, some largest particles will disappear from the aerosol due to gravitational sedimentation. Fig 7 shows the gravitational sedimentation rate of water droplets plotted against droplet sizes and shows the diffusive propagation rate for three different distances. At a distance of 0.5 m, all the aerosol particles having a diameter below 15 μm will remain in the air. Only the particles having a diameter below 7 μm will cover the 2-m distance.

By equaling diffusive propagation rate $u_d$ (Eq (7)) to the gravitational sedimentation rate, one can obtain the upper particle size limit of the aerosol which will reach distance $h$ without precipitating:

$$d_{\max} = 6\sqrt{\frac{\mu D}{hg\rho_p}}, \tag{8}$$

where $g$ is the free-fall acceleration.

Fig 8 displays maximum diameters of aerosol particles plotted against the distance from the source at two different effective coefficients of diffusion (earlier measured, $D = 1.6 \cdot 10^{-3}$ m$^2$ s$^{-1}$, and $D = 10^{-2}$ m$^2$ s$^{-1}$). The greater effective diffusion coefficient value is probably achieved when convection is more intensive in the room. Anyway, for the 2-m distance from the source,

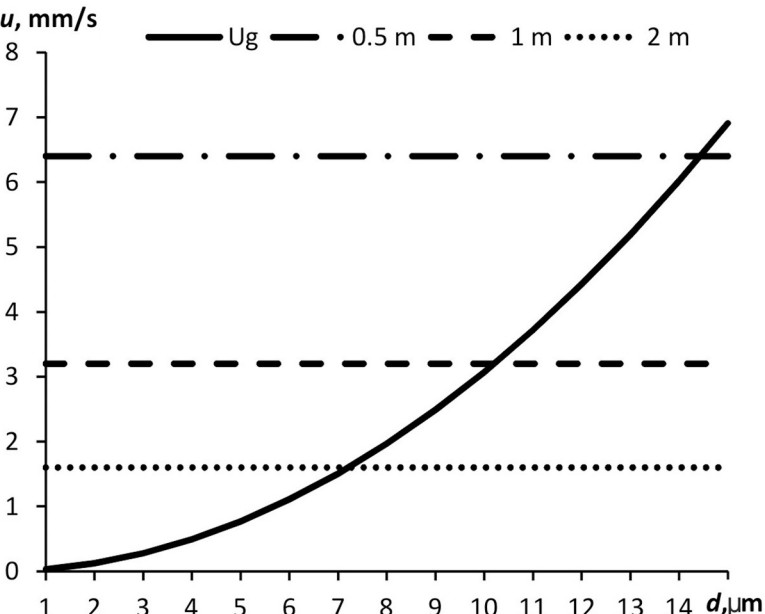

**Fig 7. Diffusive propagation rate and gravitational sedimentation rate plotted against particle diameter at different distances from the source.**

only particles having a diameter below 7 μm hover in the air. Meanwhile, according to the review by Vuorinen et al. [19] and literature cited therein, the aerosol secretions released by a human when coughing, sneezing, or talking may contain up to 87% of particles with sizes below 1 μm. Hence, even though some particles precipitate at a distance of 2 m, their major quantity will yet be hoving in the air and potentially carrying viruses.

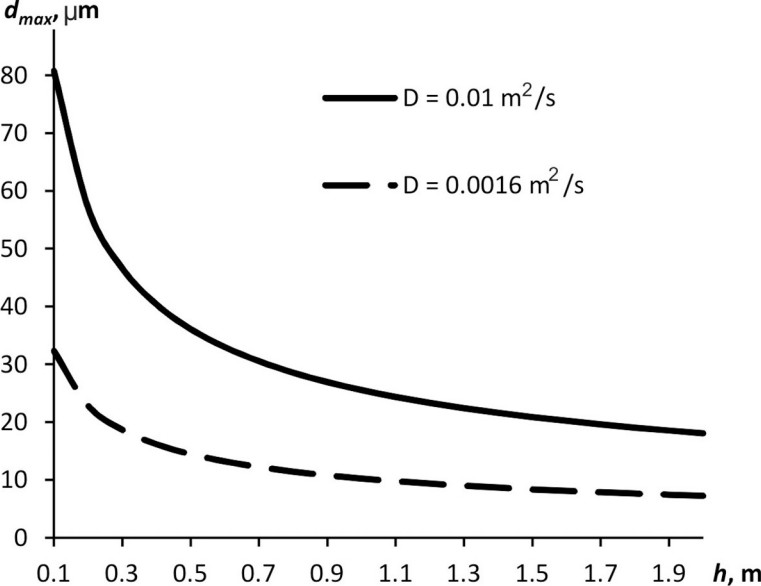

**Fig 8. Maximum aerosol particle diameter plotted against the distance from the source.**

## Discussion

As our studies demonstrate, fine aerosols propagate fast in the still air of indoor space of even complex configurations. In the case of outdoor space, this would result in a rapid decrement of aerosol concentration to safe values, whereas the particles would not disappear anywhere in indoor space and continue hoving in the air for dozens of minutes. This poses danger to the health of people around them. If an infected person as the source of bioaerosol keeps staying in the room, every subsequent act of sneezing/talking/coughing releases new portions of dangerous particles, and thereby their concentration rises. Probably, an ordinary medical face mask would not be able to diminish considerably this stream of fine particles because its mesh size is larger than that of fine particles (although such a mask is an effective means of protecting against the bulk of the larger particles of respiratory aerosol).

The intuitively unobvious deduction made from simple aerodynamic calculations suggests that the ejection velocity has no bearing on the particle propagation rate and range. The particles scatter due to diffusion processes. If that is the case (as we showed above), the propagation rate is then proportional to the effective coefficient of diffusion. In its turn, the diffusion coefficient depends on the intensity of air flows in the air. Therefore, ventilation, airing, and a running air conditioner will only increase the travel rate and range of particles.

Morawska L. et al. [25] proposes several ways to minimize the airborne transmission of Covid-19 indoors. In particular, it is necessary to enhance the efficiency of existing ventilation systems by increasing the existing ventilation rates (outdoor air change rate); to eliminate any air-recirculation within the ventilation system (just supply fresh outdoor air).

On the other hand, there is a lot of indoor air purification technologies based on air filtration and disinfection. These are mechanical HEPA filters, UV air purifiers [25] (coronavirus aerosols are 7–10 times more susceptible to ultraviolet than other viruses) [26], air extraction systems, corona discharge-generated air ions [27], household electrofilters, and adsorption technologies. We previously demonstrated [28] that electrofilters are the most efficient exactly for collecting particles less than 5 μm in size. The purification technologies are being upgraded and such appliances are getting cheaper.

Hopefully, understanding the propagation features of viral aerosols indoors will foster the advancement and application of those technologies.

## Conclusions

Here we examined the propagation processes of fine aerosols in the still air indoors. The novel Covid-19 has been found to spread via airborne and air-dust pathways. It is also known that most of the droplets resulting from sneezing, coughing, and talking have a small size (up to 16 μm). Every single sneeze or cough may release dozens of thousands of those droplets. If an infected person sneezes, coughs, or just talks, he is a steady source of the dangerous aerosol that carries viruses. We call attention to the fact that tiny particles may stay in the air, not settling on the surface for a long time. We have simulated the propagation of fine aerosols in space physically by using a pulsed aerosol generator and a model space of complex configurations. It turned out that most of the fine droplets propagate rapidly in space even in the still air and in complex geometric configurations of space. In the 3-m$^3$ test box, the particle concentration became uniform in about 3–5 min. The 1–2 m distance (a common social distance in many countries) will be covered by the aerosol particles in 5–20 min. In this case, some of them will precipitate, but most of the dangerous droplets (87%) will remain in the air. We thus suggested equations to evaluate the concentration of diffusively propagating fine particles as a function of time and distance from the source. The rates of diffusive propagation and gravitational sedimentation of different-size particles were estimated. If ventilation, air-conditioning,

airing or anything else that helps the air masses intermix intensively is used in a room, the dangerous particles will propagate faster only. The aerosol particles should be removed from the air; for that, there exist a lot of techniques that have already proved to be efficient and show prospects of further advancement.

## Supporting information

**S1 Table. Mean aerosol particles concentration vs. time.**
(DOCX)

## Acknowledgments

This work was performed using equipment of the Biysk Regional Center of Shared Use of Scientific Equipment of the SB RAS (IPCET SB RAS, Biysk city).

## Author Contributions

**Conceptualization:** Sergey S. Titov.

**Formal analysis:** Aleksandra A. Antonnikova.

**Investigation:** Aleksandra A. Antonnikova.

**Methodology:** Olga B. Kudryashova.

**Project administration:** Sergey S. Titov.

**Resources:** Evgeny V. Muravlev.

**Software:** Olga B. Kudryashova.

**Supervision:** Sergey S. Titov.

**Validation:** Aleksandra A. Antonnikova.

**Visualization:** Evgeny V. Muravlev.

**Writing – original draft:** Olga B. Kudryashova.

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
