## [Decision Letter · Decision Letter 0]

9 Nov 2020

PONE-D-20-30533

Propagation of viral bioaerosols indoors

PLOS ONE

Dear Dr. Kudryashova,

Thank you for submitting your manuscript to PLOS ONE. After careful consideration, we feel that it has merit but does not fully meet PLOS ONE’s publication criteria as it currently stands. Therefore, we invite you to submit a revised version of the manuscript that addresses the points raised during the review process.

We look forward to receiving your revised manuscript.

Kind regards,

Amitava Mukherjee, ME, Ph.D.

Academic Editor

PLOS ONE

Journal Requirements:

2.Thank you for stating the following in the Acknowledgments Section of your manuscript:

[This work was performed using equipment of the Biysk Regional Center of Shared

Use of Scientific Equipment of the SB RAS (IPCET SB RAS, Biysk city).]

 [The author(s) received no specific funding for this work.]

Reviewers' comments:

Reviewer's Responses to Questions

**Comments to the Author**

1. Is the manuscript technically sound, and do the data support the conclusions?

Reviewer #1: Partly

Reviewer #2: Yes

2. Has the statistical analysis been performed appropriately and rigorously? 

Reviewer #1: No

Reviewer #2: Yes

3. Have the authors made all data underlying the findings in their manuscript fully available?

Reviewer #1: Yes

Reviewer #2: Yes

4. Is the manuscript presented in an intelligible fashion and written in standard English?

Reviewer #1: Yes

Reviewer #2: Yes

5. Review Comments to the Author

Reviewer #1: In this study, the authors have aimed to assess how virus particles may propagate in the indoor aerosol, using a model experiment and established mathematical equations. Although the experiment has correctly been designed, a reader cannot find any statistical data. It looks like a single model experiment; hence, it is not clear whether the data can be treated as valid. Nevertheless, the results are original and the simulation of virus propagation via aerosol is creative. Besides, some improvements are required in the text. Firstly, the study aims might be based on hypotheses that are verified by the experimental data. It might be clear what they have wanted to test using the model proposed in the study. Secondly, the authors should provide proves (references) for some statements to avoid speculation (l. 16-17, p.3; l. 1-2, p. 4). Thirdly, they are advised to use degrees Celsius for temperature data (p. 3). Fourthly, COVID-19 is the name of a pandemic; the name of the virus is SARS-CoV-2 (abstract). Fourthly, the reader can question the statement that plenty of RNA fragments of the coronavirus was detected in Wuhan hospitals. Can be regarded 4, 24, 30 copies per cubic meter of the indoor air as plenty of viruses? Maybe the increased numbers of virus particles would be a better statement. You do not know any limits of the SARS-CoV-2 in the indoor air that can be treated as low, medium or high. Finally, the statement that an ordinary medical mask would not be able to diminish considerably the stream of fine virus particles because of its mesh size can be questioned by the reader (p. 13). Several authors have published articles showing 80% effectiveness of the surgical masks in blocking the transmission of expelled virus particles by man. Even simple cotton masks have shown approx. 60% efficacy (e.g., Disaster Medicine and Public Health Preparedness 7(4):313-218 DOI:10.1017/dmp.2013.43). The authors can check information included in the portal https://smartairfilters.com/en/blog and cited references there. Furthermore, the sentence on page 14 might be corrected for better style (i.e. “Morawska et al. [23] proposed several ways to minimize…”).

Reviewer #2: Review comments

Thank you for providing me the opportunity to review the article titled “Propagation of viral bioaerosols indoors”. The manuscript deals with the spread of aerosols indoors that carry viruses. Overall, the manuscript conveys an important viewpoint and have conveyed that to an extent. The comments and suggestion mentioned below can improve the scientific value of the manuscript and prepare it for publication.

General comments:

First of all, the authors did not include line numbering in the manuscript which creates difficulties for reviewers to comment. The authors are recommended to check the guide for authors.

The manuscript is well written in most of the areas, however a thorough check for language errors by native speakers can improve the readability in some areas.

For example:

• Rewrite the first sentence in introduction section (The coronavirus pandemic…);

• Rewrite the second sentence in 2nd paragraph of Materials section (Under low humidity…);

• Check the spellings of the words in the 2nd paragraph of the Primary scatter and formation time of fine aerosol cloud section (maxium, miliseconds);

• Check the spellings of the words in the 3rd paragraph of Assessment of characteristic velocities of processes (dissapear; dimeter; dimateer; esarlier);

• Check the word order in the sentence in the Conclusion section (We simulated physically…).

Check similar errors in each section and try to minimize them.

Specific comments:

The authors used different ways of expressing numbers in their manuscript. For example, in the section of On the effective coefficient of diffusion, Db = 10-12 m2s-1 (scientific notation) and D = 0.0016 m2s-1 (decimal notation). Try to use the same format in the manuscript.

In the introduction section, the authors stated that viruses were found mostly in particles less than 5µm in diameter. However, is it logical to conduct a physical simulation using particles up to 20 µm. Studies found that aerosols less than 1µm (PM1) can carry viruses (Baron, P. (2010). Generation and behavior of airborne particles (aerosols)). This part needs more clarity.

In the section of Initial data for simulation, the authors mentioned the initial ejection velocity of droplets as 35m/s. However, in the section of Primary scatter and formation time of fine aerosol cloud, 300 m/s was mentioned as an initial particle velocity. These parts need more clarity.

Include some references related to particles settlement and toxicity of finer particles to support your point. Check these articles:

Baron, P. (2010). Generation and behavior of airborne particles (aerosols)

Tian, G., Wang, J., Lu, Z., Wang, H., Zhang, W., Ding, W., & Zhang, F. (2019). Indirect effect of PM1 on endothelial cells via inducing the release of respiratory inflammatory cytokines. Toxicology in Vitro, 57, 203-210.

6. PLOS authors have the option to publish the peer review history of their article (what does this mean?). If published, this will include your full peer review and any attached files.

Reviewer #1: No

Reviewer #2: No

---

## [Author Response · Author response to Decision Letter 0]

17 Nov 2020

The authors are grateful to the Reviewers for their careful reading of the manuscript and useful comments. We hope that the manuscript will get better thanks to these comments.

Reviewer #1

1. In this study, the authors have aimed to assess how virus particles may propagate in the indoor aerosol, using a model experiment and established mathematical equations. Although the experiment has correctly been designed, a reader cannot find any statistical data. It looks like a single model experiment; hence, it is not clear whether the data can be treated as valid. Nevertheless, the results are original and the simulation of virus propagation via aerosol is creative. Besides, some improvements are required in the text.

We have added S1.Table (Supplementary Materials), which contains the results and statistics of all experiments presented in the manuscript. The confidence interval does not exceed 8%.

2. Firstly, the study aims might be based on hypotheses that are verified by the experimental data. It might be clear what they have wanted to test using the model proposed in the study.

At the end of the Introduction, we added the phrase: 

“We are going demonstrate experimentally and theoretically that a lot of fine droplets released from an infected person’s coughing, sneezing, or talking propagate very fast and for large distances indoors, as well as bend around obstacles, lift up and down over staircases, and so on.”

3. Secondly, the authors should provide proves (references) for some statements to avoid speculation (l. 16-17, p.3; l. 1-2, p. 4). 

We've added a reference:

5. Tian G, Wang J, Lu Z, Wang H, Zhang W, Ding W, Zhang F. Indirect effect of PM1 on endothelial cells via inducing the release of respiratory inflammatory cytokines. Toxicol. in Vitro, 2019;57: 203–210. Available from: https://doi.org/10.1016/j.tiv.2019.03.013

18. Baron P. Generation and behavior of airborne particles (Aerosols). Division of Applied Technology. National Institute for Occupational Safety and Health, Centers for Disease Control and Prevention. 2010.

4. Thirdly, they are advised to use degrees Celsius for temperature data (p. 3). 

Corrected.

5. Fourthly, COVID-19 is the name of a pandemic; the name of the virus is SARS-CoV-2 (abstract). 

Corrected.

6. Fourthly, the reader can question the statement that plenty of RNA fragments of the coronavirus was detected in Wuhan hospitals. Can be regarded 4, 24, 30 copies per cubic meter of the indoor air as plenty of viruses? Maybe the increased numbers of virus particles would be a better statement. You do not know any limits of the SARS-CoV-2 in the indoor air that can be treated as low, medium or high. 

Yes, this is true, we do not know whether these are many viruses or not. We supplemented the second paragraph of the Introduction with a sentence:

“Perhaps the concentration of viral particles is not so high as to cause a danger of infection, but the very fact of the transfer of viral particles by small aerosol droplets far from the source of origin is interesting.”

7. Finally, the statement that an ordinary medical mask would not be able to diminish considerably the stream of fine virus particles because of its mesh size can be questioned by the reader (p. 13). Several authors have published articles showing 80% effectiveness of the surgical masks in blocking the transmission of expelled virus particles by man. Even simple cotton masks have shown approx. 60% efficacy (e.g., Disaster Medicine and Public Health Preparedness 7(4):313-218 DOI:10.1017/dmp.2013.43). The authors can check information included in the portal https://smartairfilters.com/en/blog and cited references there.

Yes, we are not arguing about the effectiveness of medical masks. They actually reduce the flow of aerosol particles by up to 90% by weight. However, they do not retain up to 90% of the number of small aerosol particles that carry viruses. Probably, such particles are not dangerous in low concentrations, so masks are indeed an effective means of protection (for example, when shopping). But with a long stay in a room with a sick person, even if you and he are wearing a mask, it can be dangerous, since the concentration of small particles constantly increases and at some point can reach a critical value (which we do not know, this is the subject of separate studies ). Actually, this is what the article is about.

The proposal now looks like this: 

“Probably, an ordinary medical face mask would not be able to diminish considerably this stream of fine particles because its mesh size is larger than that of fine particles (although such a mask is an effective means of protecting against the bulk of the larger particles of respiratory aerosol).”

8. Furthermore, the sentence on page 14 might be corrected for better style (i.e. “Morawska et al. [23] proposed several ways to minimize…”).

Corrected.

Reviewer #2

1. First of all, the authors did not include line numbering in the manuscript which creates difficulties for reviewers to comment. The authors are recommended to check the guide for authors.

We numbered the lines

2. The manuscript is well written in most of the areas, however a thorough check for language errors by native speakers can improve the readability in some areas.

For example:

• Rewrite the first sentence in introduction section (The coronavirus pandemic…);

It was: “The coronavirus pandemic that stroke the world in 2019 has raised many concerns before human kind, which have to be addressed.”

Now is: “The coronavirus pandemic that stroke the world in 2020 has posed a series of questions for humanity that need to be solved.”

• Rewrite the second sentence in 2nd paragraph of Materials section (Under low humidity…);

It was: “Under low humidity conditions typical of rooms, the said process ends rapidly, and only non-volatile residues of droplets are now involved in a slower diffusive propagation stage.” 

Now is: “Under low humidity conditions typical of indoor environments, the evaporation stage ends rapidly. This is followed by a slower stage of the diffusive propagation of non-volatile residues of droplets.”

• Check the spellings of the words in the 2nd paragraph of the Primary scatter and formation time of fine aerosol cloud section (maxium, miliseconds);

Corrected: “Maximum distance R is linearly dependent on initial velocity u (Fig 4). The time, over which the maximum distance is covered, is estimated at several milliseconds.”

• Check the spellings of the words in the 3rd paragraph of Assessment of characteristic velocities of processes (dissapear; dimeter; dimateer; esarlier);

Corrected: “disappear, diameter, earlier”

• Check the word order in the sentence in the Conclusion section (We simulated physically…).

Corrected. It was: “We simulated physically the propagation of fine aerosols in space by using a pulsed aerosol generator and a model space of complex configurations.”

Now is: “We have simulated the propagation of fine aerosols in space physically by using a pulsed aerosol generator and a model space of complex configurations.”

3. Check similar errors in each section and try to minimize them.

We've fixed a lot of minor bugs like missing letter in a word or comma.

Specific comments:

4. The authors used different ways of expressing numbers in their manuscript. For example, in the section of On the effective coefficient of diffusion, Db = 10-12 m2s-1 (scientific notation) and D = 0.0016 m2s-1 (decimal notation). Try to use the same format in the manuscript.

Fixed. For instance: D = 1.6•10-3 m2 s-1, and D = 10-2 m2 s-1

5. In the introduction section, the authors stated that viruses were found mostly in particles less than 5µm in diameter. However, is it logical to conduct a physical simulation using particles up to 20 µm. Studies found that aerosols less than 1µm (PM1) can carry viruses (Baron, P. (2010). Generation and behavior of airborne particles (aerosols)). This part needs more clarity.

In the introductory part, we referred to works that indicate the typical droplet size for sneezing and coughing (up to about 16 microns). The relatively large droplets are likely to be trapped by the mask, or settle under the influence of gravity at a "social" distance. On the other hand, particles less than 5 microns in diameter still carry dangerous portions of viruses. And it is they who fly away from the source. But it is nevertheless necessary to simulate an aerosol with a full spectrum of particle sizes, approximately corresponding to a cough-sneeze. Which is what we did. Thanks for the reference, we added it!

6. In the section of Initial data for simulation, the authors mentioned the initial ejection velocity of droplets as 35m/s. However, in the section of Primary scatter and formation time of fine aerosol cloud, 300 m/s was mentioned as an initial particle velocity. These parts need more clarity.

As follows from Figures 3 and 4, particles at an initial velocity of 35 m/s and at 300 m/s will stop near the source (3 cm and 20 cm, respectively). And their further propagation will be determined by the diffusion mechanism. Therefore, for the problem of describing the propagation of particles over distances of several meters, the difference in the initial velocity is not important. We added the phrase:

“At an ejection velocity of u = 300 m/s small particles will stop at a distance about 20 cm from the source. Thus, that difference in initial velocity is not important for describing the propagation of particles over distances of several meters.”

7. Include some references related to particles settlement and toxicity of finer particles to support your point.

Thank you, we’ve added them ([5, 18]). (Baron's presentation is impressive).

---

## [Decision Letter · Decision Letter 1]

21 Dec 2020

Propagation of viral bioaerosols indoors

PONE-D-20-30533R1

Dear Dr. Kudryashova,

We’re pleased to inform you that your manuscript has been judged scientifically suitable for publication and will be formally accepted for publication once it meets all outstanding technical requirements.

Kind regards,

Amitava Mukherjee, ME, Ph.D.

Academic Editor

PLOS ONE

Additional Editor Comments (optional):

Reviewers' comments:

Reviewer's Responses to Questions

**Comments to the Author**

1. If the authors have adequately addressed your comments raised in a previous round of review and you feel that this manuscript is now acceptable for publication, you may indicate that here to bypass the “Comments to the Author” section, enter your conflict of interest statement in the “Confidential to Editor” section, and submit your "Accept" recommendation.

Reviewer #1: All comments have been addressed

2. Is the manuscript technically sound, and do the data support the conclusions?

Reviewer #1: Yes

3. Has the statistical analysis been performed appropriately and rigorously? 

Reviewer #1: Yes

4. Have the authors made all data underlying the findings in their manuscript fully available?

Reviewer #1: Yes

5. Is the manuscript presented in an intelligible fashion and written in standard English?

Reviewer #1: Yes

6. Review Comments to the Author

Reviewer #1: The manuscript has been revised according to my comments. I do not require further improvements in the text.

7. PLOS authors have the option to publish the peer review history of their article (what does this mean?). If published, this will include your full peer review and any attached files.

Reviewer #1: No

---

## [Editor Report · Acceptance letter]

23 Dec 2020

PONE-D-20-30533R1 

Propagation of viral bioaerosols indoors 

Dear Dr. Kudryashova:

I'm pleased to inform you that your manuscript has been deemed suitable for publication in PLOS ONE. Congratulations! Your manuscript is now with our production department. 

Kind regards, 

on behalf of

Professor Dr. Amitava Mukherjee 

Academic Editor

PLOS ONE